# Perceptual judgments are resistant to the advisor's perceived level of trustworthiness: A deep fake approach

**Mathias Van der Biest**[1]*, **Sam Verschooren**[1,2,3], **Frederick Verbruggen**[1,4], **Marcel Brass**[1,2]*

1 Department of Experimental Psychology, Faculty of Psychology and Educational Sciences, Ghent University, Ghent, Belgium, 2 Department of Psychology, Berlin School of Mind and Brain, Humboldt-Universität zu Berlin, Berlin, Germany, 3 Max Planck Institute for Human Brain and Cognitive Sciences, Leipzig, Germany, 4 Department of Biology, Faculty of Sciences, Ghent University, Ghent, Belgium

* marcel.brass@hu-berlin.de (MB); Mathias.VanderBiest@UGent.be (MVDB)

## Abstract

As we navigate our environment, we frequently make spontaneous judgments about other's characteristics. Trustworthiness is a particularly important trait, often judged instantly and used to guide decisions, especially in uncertain situations. Although the impact of trustworthiness on social behaviour is well-documented, its influence on more fundamental cognitive processes, such as perceptual decision-making, remains unclear. The present study aims to fill this gap. In the first experiment (N = 100), we validated a new trustworthiness manipulation by applying deep fake technology to create animated versions of perceptually trustworthy, untrustworthy, and neutral static computer-generated faces. In the second experiment (N = 199), the deep fake procedure was applied to a new set of trustworthy and untrustworthy faces that served as advisors during a perceptual decision-making task. Here participants had to indicate the direction of dots that were either moving coherently to the left or the right (i.e., random dot motion task). Contrary to our predictions, participants did not align more with the advice of trustworthy advisors than that of untrustworthy advisors. While participants made faster decisions and reported greater confidence when aligning with the advice, these effects were not influenced by the advisor's perceived trustworthiness. We integrate our findings within theoretical frameworks of advice taking, domain specificity of facial trustworthiness, and task requirements.

## Introduction

When making decisions, social beings like humans, often rely on others [1], particularly in uncertain situations. In such cases, understanding the intentions of interaction partners becomes critical for determining whether to engage with or disregard the information they provide [2,3]. Social cues, such as perceived facial trustworthiness, play a key role in inferring these intentions [4–7]. Trustworthiness forms the basis of establishing meaningful relationships [8,9], and our cooperative behaviour [10]. It is a trait that we spontaneously [4] and rapidly evaluate [11,12], that tends to remain stable over time [13], and is consistent across age groups, from toddlers to the elderly [14–16]. Crucially, these trustworthiness evaluations

**Data availability statement:** All presented results are based on new data, collected by the authors. All data and scripts can be found here: https://osf.io/bu7dj/?view_only=5e94356b2d-554f17b1125cdf9c8468a5

**Funding:** M.V.D.B. (11K2721N) and S.V (1212721N) are supported by the Research Foundation Flanders. F. V is supported by an ERC Consolidator grant (European Union's Horizon 2020 research and innovation programme, Grant Agreement No 769595). M.B. is supported by an Einstein Strategic Professorship of the Einstein Foundation Berlin (EPP-2018-483) and by the Deutsche Forschungsgemeinschaft (DFG, German Research Foundation) under Germany's Excellence Strategy–EXC 2002/1 'Science of Intelligence'–project number 390523135. The funders had no role in the study design, data collection and analysis, decision to publish, or preparation of the manuscript.

**Competing interests:** The authors have declared that no competing interests exist.

influence our behavior. For example, participants invested more money [17], followed advice more and faster [18], and made more risky choices when their interaction partner was perceived as trustworthy [19]. Similarly, individuals with an untrustworthy appearance were perceived as more likely to have committed a crime [20], were convicted more quickly [21], and were remembered better [22]. Such social biases have been primarily demonstrated in social (e.g., criminal judgments), motivational (e.g., cooperation), economic (e.g., investments), and value-based decision-making contexts. While these findings highlight trustworthiness as a critical factor in decision-making, its influence on lower-level sensory processes, such as perceptual decision-making, remains unexplored. To address this gap, we investigated the impact of perceptual trustworthiness on decision-making during a random dot motion task.

Perceptual decision-making plays a fundamental role in helping individuals navigate and interact with the world around them [23]. For example, we rely on it to judge whether a car is far away or nearby when crossing a road or to decide whether food on our plate is fresh or mouldy. In laboratory settings, perceptual decision-making is often assessed using visual discrimination tasks, such as the random dot motion task, which participants frequently perform individually [24]. However, in everyday life, (perceptual) decision making often occurs in interaction with others [25]. On one hand, integrating information from multiple sources has been shown to enhance task performance [26,27], facilitate efficient problem solving [28], improve motor performance [29], and increase the identification of misinformation [30], making it a key aspect of group behavior [1]. This applies to perceptual decisions as well, for example, studies have shown that accuracy in perceptual decision tasks depends on the accuracy of task partners [31]. Moreover, exchanging task-related confidence ratings can improve accuracy on perceptual tasks [32], and interaction is crucial when task partners differ in perceptual accuracy to achieve proficient performance [33]. On the other hand, evidence from the advice taking literature indicates that decision-makers often overestimate their own judgments while underweighting the input of others [34–36]. When given the choice to base decisions on group information or act individually, participants frequently ignored group input, even when doing so led to more errors [36], and when combining information from different sources was statistically advantageous [37].

Thus, advice from others can play a crucial role in decision-making. However, not all advice is used equally. For example, people tend to prioritise advice from experts over that from novices [38]. More broadly, advice taking and information integration depend on the characteristics of the interaction partner [34,39] and on the level of uncertainty. Social influence theories [40] and empirical studies [41,42] propose that we are more susceptible to social influences when we are uncertain and lack confidence in our abilities or decisions [43–45]. For example, during perceptual decision tasks, ambiguous sensory information increases reliance on collaborator's reputations compared to tasks with unambiguous information [46]. Participants were also more likely to seek additional information from advisors with unshared information, perceiving it as a sign of greater knowledge [47]. Other studies have found that participants were less likely to follow other's advice when confident in their own decisions [48], and individuals with greater self-assurance were less inclined to rely on social cues [49].

Overall individuals tend to integrate additional information when making decisions in uncertain situations. Social cues, such as perceived trustworthiness [7] are then used to evaluate the quality of the information [50], to make a more informed decisions. From an evolutionary perspective, fast and often spontaneous trustworthiness impressions [12,13] help us decide whom to approach and whom to avoid [4,6,7]. These impressions, based on perceptual information, could then provide an intuitively accessible source of information that acts as a

heuristic for navigating our environment [51]. For example, a recent study found that people are more likely to follow the advice of perceived trustworthy advisors, and do so more quickly, when making value-based decisions—a type of subjective decision-making where choices depend on personal opinions or the perceived value of the options [18]. This trustworthiness advice following bias is in line with findings using non-perceptual trustworthiness manipulations such as the validity of the advisor (i.e., trustworthy; highly rewarding advice, untrustworthy: highly unrewarding advice) [52,53]. Trustworthiness is thus an important predictor of advice utilisation [18,39], particularly in subjective decision-making contexts lacking an objectively correct answer.

To investigate the impact of perceptual trustworthiness on advice-taking in a perceptual decision-making context, we first selected a series of static computer-generated faces previously rated as trustworthy, neutral, or untrustworthy [4]. Using a novel deep fake technique [54], we transformed these static images into interactive, dynamic social partners. This approach reflects the dynamic nature of real-life interactions and enhances the ecological validity of our paradigm by incorporating dynamic faces capable of providing verbal advice. In Experiment 1 (N = 100), we validated our deep fake procedure and assessed whether these animated faces still elicit differences in their perceived trustworthiness. Next, we applied the same deep fake procedure to create a new set of trustworthy and untrustworthy game partners. These animated faces advised participants during a random dot motion task. In this task, participants were presented with a cloud of coherent moving (i.e., moving in the same direction) and incoherent moving dots (i.e., moving in different directions) and were required to indicate the direction of the coherent movements by pressing left or right. The task had two levels of difficulty, namely hard trials with a coherency of 5–10% (i.e., 55–60% of the dots are moving in one direction) or easy trials with a coherency of 25% (i.e., 75% of the dots are moving the one direction). For the easy trials, the advice was always correct, for the hard trials, the advice was sometimes correct (78%) and sometimes incorrect (22%). The frequency and speed (i.e., choice decision time) with which the participants decisions aligned (i.e., advice alignment rate) with the game partner's advice were measured. Additionally, we included a confidence rating after each trial to measure the level of (un)certainty [55–57]. After each trial, participants indicated how confident they were about their decision on a continuous scale (0–100).

Consistent with previous studies on value-based decision-making, we hypothesised that on hard trials with correct advice, participants would respond faster and their decisions would align more with the advice of trustworthy advisors than with that of untrustworthy advisors [18,53]. We expected participants to be more confident in their decisions when the advisor was perceived as trustworthy Fcompared to untrustworthy. In addition to the main effects of trustworthiness, we predicted interaction effects between trustworthiness and advice alignment for the confidence ratings and the choice decision time (i.e., how fast do participants decide). The interaction for confidence ratings was based on the findings that the advisor's trustworthiness influences advice discounting and utilisation [53] and that integrating information from others, such as in an advice contexts, results in a confidence boost [58]. It was therefore predicted that participants would be the most confident about the decision when the advisor was trustworthy and their decision aligned with the advice, and the least confident when their decision did not align with the advice, and the advisor was trustworthy. The same logic can be applied to the choice decision time. Previous research showed that participants decide faster when they followed advice as well as when they received advice from a perceived trustworthy advisor [18]. Therefore, the largest difference was expected between the advice alignment rate for the perceived trustworthy advisors compared to the perceived untrustworthy advisors.

## Method experiment 1

The validation study aimed to investigate whether deep fake manipulations [54] of static computer generated faces that were previously categorised as trustworthy, untrustworthy, and neutral [4] would maintain their trustworthiness levels when animated.

### Participants

Data from 110 participants ($M_{age}$ = 32, $SD_{age}$ = 9, 87 female, 22 male, 1 not reported) on Prolific [59] were collected. Only participants with English as their native language, an approval rate of at least 70% based on at least 10 submissions on Prolific (www.prolific.com), and no prior participation in our previous studies were allowed. Participants who responded too quickly (N = 4, 1.5IQR from the 25th percentile), only used one response (N = 1), or made more than 50% mistakes on our attention check (N = 4) were excluded. Furthermore, one participant was excluded due to multiple exclusion criteria. As a result, the total sample size was 100 ($M_{age}$ = 32, $SD_{age}$ = 10, 79 females, 20 males, 1 not reported).

In addition to the recruited participants, 52 participants returned the experiment before completing it and 5 timed out. The median completion time was 14.30 minutes, and participants were paid at a rate of 5£ an hour.

The experiment and applied procedures were conducted according to the guidelines of the ethical committee of Ghent University (approval number: 2020/167).

### Apparatus and materials

To create animated versions of static images, deep fake models require two inputs: a source video to provide the animation and a static image to be animated [54]. For the source videos, 40 male individuals ($M_{age}$ = 41, $SD_{age}$ = 14) with English as their native language were recruited through Prolific. Each participant was requested to introduce themselves in a video (e.g., 'Hi, my name is Mark'). The names were generated using an English name generator (Reedsy). We randomly selected 120 computer-generated faces from the Oosterhof and Todorov database [4], created with FaceGen (www.facegen.com), representing three categories: trustworthy (+3 SD), neutral (0 SD), and untrustworthy (-3 SD), with 40 faces per category. Since deep fake models can only animate static images without incorporating the audio from the source video, the sound was added to the animations after generating the deep fakes. To minimise the impact of the voice on our trust ratings, each voice was used for each level of trustworthiness. Finally, the videos were centred in a 256*256 square with a black background for perceptual similarity purposes. You can find some examples of the deep fakes on OSF (OSF).

The experiment was conducted using Google Chrome in full-screen mode and programmed in Javascript with JsPsych [v6.0.5, 60]. Participants were presented with deep fake videos in a random order and asked to rate the trustworthiness of each virtual character on a scale of 1 (very untrustworthy) to 9 (very trustworthy) using their computer mouse.

### Procedure

Participants provided their consent before taking part by pressing the consent button with their computer mouse. In each trial, they viewed a deep fake video lasting between 1 and 3 seconds, in which the avatar introduced themselves with the phrase 'Hi, my name is X'. After viewing the video, participants selected the 'continue' button and were then asked to rate the virtual character's trustworthiness on a Likert scale ranging from 1 to 9. The order of the videos was randomized. Additionally, participants were required to listen carefully to the names of the virtual faces. On 10 random trials, they were asked to type in the name of the avatar

they had just been introduced to. This was used as an attention check and was always presented after participants had rated the avatar.

## Design

We had a within-subject design with one factor consisting of three levels (trustworthiness: trustworthy, neutral, and untrustworthy).

## Preprocessing and analyses

Preprocessing was done in R [v4.3.1, 61], and analyses were conducted in JASP [v0.18.3, 62]. All trials that deviated by more than 2 standard deviations from the participant's decision times means were considered outliers (4%). To centre the data around zero, we subtracted the variable mean from each data point. Additionally, we standardised the data variability by dividing the data points by the standard deviation. Both centring and scaling were performed using the scale function (v 4.3.1, base Package).

The data was analysed using a within-subject repeated measures ANOVA (Type III) [63,64] with trustworthiness as a factor. Post hoc t-tests were conducted to determine the direction of significant effects, and the corresponding p-values were corrected using Holm's correction [65]. Lastly, we ran a one sample t-test, to see if each level of trustworthiness was different from zero, which was set as the reference level due to our centring and scaling approach. Additionally, equivalent Bayesian analyses were performed using the default settings of JASP [v0.18.3, 62], and interpreted according to the JASP guidelines [66].

## Results

The assumption of sphericity was violated, as confirmed by Mauchy's test for sphericity, $W = 0.59$, $\chi^2_2 = 51.09$, $p < .0001$. Therefore, the reported results were corrected using the Greenhouse-Geisser correction. Our analyses showed that participants significantly rated the trustworthy faces as the most trustworthy (M = 0.16, SD = 0.47), the untrustworthy as the least trustworthy (M = −0.20, SD = 0.52), and the neutral faces in between the trustworthy and untrustworthy ratings (M = 0.05, SD = 0.48), $F(1.14, 140.80) = 93.00$, $MSE = 0.050$, $p < .0001$, $\eta^2_p = 0.48$ (see Fig 1). The post hoc analyses revealed that the trustworthy faces received significantly higher trustworthiness ratings than the neutral and untrustworthy faces ($t(99) = 3.93$, $p < .00001$, $d = 0.22$, $BF_{10} = 31246$; $t(99) = 13.28$, $p < .00001$, d = 0.74, $BF_{10} > 1000000$, respectively). Additionally, the neutral faces were rated as more trustworthy than the untrustworthy faces, $t(99) = 9.34$, $p < .00001$, $d = 0.52$, $BF_{10} > 1000000$.

We conducted a one-sample t-test to compare each level of trustworthiness to the centred level (0). The trustworthy faces received significantly higher ratings than zero ($t(99) = 3.39$, $p = .001$, $d = 0.33$, $BF_{10} = 19$), while the untrustworthy faces received significantly lower ratings than zero ($t(99) = −3.93$, $p < .001$, $d = −0.39$, $BF_{10} = 120$). However, the neutral faces did not significantly differ from zero ($t(99) = 1.06$, $p = .29$, $d = 0.11$, $BF_{01} = 5.26$).

## Discussion experiment 1

In our validation study, we demonstrated that after applying the deep fake procedure to transform the static computer-generated images into a video format, animated trustworthy faces were rated as trustworthy, neutral faces as neutral, and untrustworthy faces as untrustworthy This was confirmed by both frequentist and Bayesian analyses. Additionally, the centred and scaled data for the neutral faces were not significantly different from zero; by contrast, the trustworthy (larger than zero) and untrustworthy (smaller than zero) faces were significantly different from the baseline (i.e., zero). Overall, the experiment demonstrated the success of our deep fake procedure.

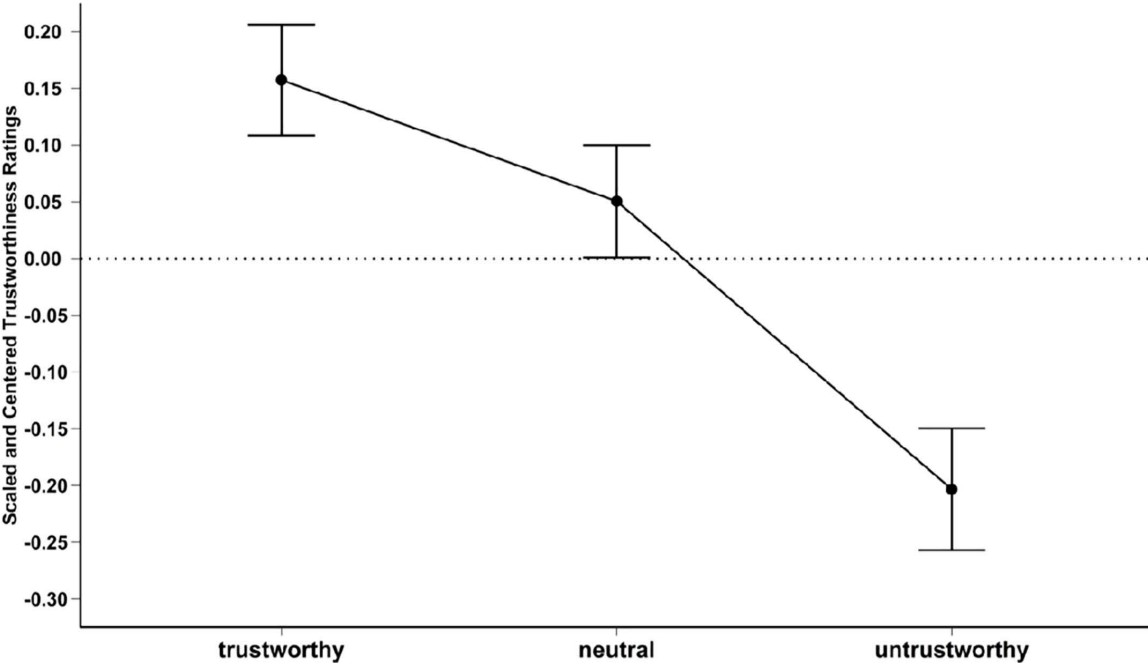

**Fig 1. Results of the trustworthiness validation study.** On the x-axis you can find the different levels of trustworthiness (trustworthy, neutral, and untrustworthy), and on they y-axis the scaled and centred trustworthiness ratings. The black dots represent the mean of each level of trustworthiness and the black bars represent the error bars.

## Method experiment 2

Given the conclusive results of our validation study, the same procedure was applied when creating the animated faces used in our main experiment without rerating them. In this pre-registered experiment (preregistration), we investigated whether perceived trustworthiness, as manipulated with our deep fake animations, modulates perceptual decision making.

### Participants

We recruited 244 participants ($M_{age}$ = 30.8, $SD_{age}$ = 5.75, 124 female, 119 male, and 2 who did not report their gender) on Prolific [59]. Participants were required to have at least 10 submissions on Prolific (www.prolific.com), be native English speakers between the ages of 18 and 40 and could not have participated in any of our previous studies. Participants whose accuracy was below or equal to 66% on the easy coherency trials (N = 23), those with an accuracy below or equal to 75% on our attention check (N = 31), or those who responded faster or slower than 1.5 interquartile range from the 25th or 75th percentile (N = 2), were excluded from the experiment. Of the excluded participants, 10 violated multiple exclusion criteria. The final sample size was 199 ($M_{age}$ = 30.70, $SD_{age}$ = 5.77, with 98 females and 100 males; one participant did not report their gender). We did not deviate from our preregistration.

The sample size was determined through a power analysis using G*power [67] before the experiment. Since it is currently not possible to calculate the sample size of (general) linear mixed models with G*power, a classic repeated measures ANOVA design was assumed with a small effect size (f = .10), power of .80, and an alpha level of .05. Although we recognize that this sample size determination approach is suboptimal for (general) linear mixed models

compared to simulations [68], we reasoned that the applied method is a more conservative estimate of the sample size; after all, linear mixed models account for more unexplained variance as they take both fixed and random effects into account.

In addition to the recruited participants, 73 participants started the experiment but either did not complete it on time or returned their submission. The median completion time was 21.15 minutes, and participants were compensated at a rate of £6 per hour, in accordance with Prolific's guidelines.

The experiment and applied procedures were conducted according to the guidelines of the ethical committee of Ghent University (approval number: 2024-084W).

## Apparatus and materials

For the trustworthiness stimulus, we applied the same deep fake procedure used in our validation study to four randomly selected faces (i.e., +3SD four trustworthy, and -3SD four untrustworthy) from [4]. This database consists of 100 face identities with three distinct levels trustworthiness (e.g., untrustworthy, neutral, and untrustworthy) for each face identity (i.e., 300 faces in total). As we wanted to assure that each face consisted of a different face identity, we selected four new faces with a different face identity for each level of trustworthiness (i.e., trustworthy, untrustworthy). The source video used was of the first author of this paper saying 'left' or 'right'. We combined this with the voices of four UK based male English native speakers, which were recorded in a previous experiment [69]. The audio was edited onto the deep fake videos afterwards. The deep fakes were sized 480 by 480 pixels and presented on a black background, serving as the advisor during our experiment.

The random dot motion task was programmed in Javascript using the JsPsych Library [60] and could only be run on Google Chrome. The 'ROK' plugin [70] was used to generate the random dot motion trials. The stimulus consisted of 100 dots presented in an ellipse that moved either to the left or right, with a speed setting in ROK of 40 (i.e., percentage of width ellipse/seconds). The size of the ellipse, dots, and moving distance were scaled according to the participants' screen size. This rescaling was also applied to the deep fakes and static images. The task had two levels of difficulty, namely hard trials with a coherency of 5–10% or easy trials easy trials with a coherency of 25%. For the easy trials, the advice was always correct, for the hard trials, the advice was sometimes correct (78%) and sometimes incorrect (22%). This resulted in three trial types (i.e., easy, hard-correct; hard-incorrect). This distinction between hard and easy trials was based on an earlier study, which found stronger social bias effects in hard compared to easy trials [46]. In addition, the differences in validity were introduced to mimic the accuracy of the participants. This to increase the ecological validity of the advice context. Participants were required to indicate the direction of the coherent dots by pressing 'c' for left and 'n' for right. For the confidence ratings, participants indicated their level of confidence by moving a cursor on a continuous scale from 0 (not very confident) to 100 (very confident). The cursor always started in the middle and participants had to move their mouse before continuing.

The experiment was conducted in full-screen mode with a black background.

## Procedure

Participants agreed with the informed consent form before starting the experiment by pressing the consent button with their computer mouse. An overview of the task was presented, and the participants received the task instructions for each phase. This modified random dot motion paradigm consisted of three phases: an introduction, a random dot motion, and a confidence rating phase (see Fig 2).

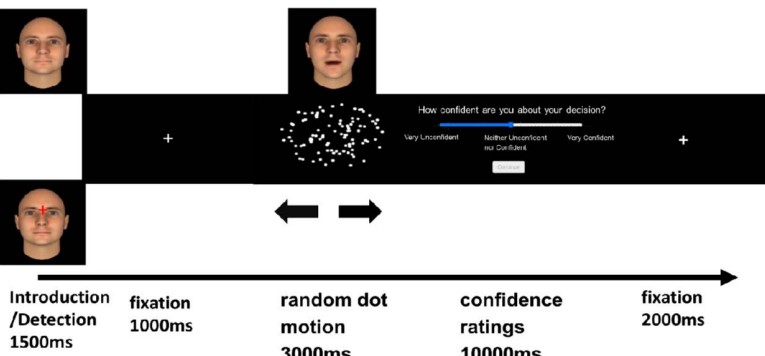

**Fig 2. One trial of the experimental procedure.** In the figure, an example trial of our experimental procedure is presented. Participants were introduced to an advisor with whom they will play in the current trial. On some trials, a red cross was displayed, and participants were required to press the space bar as quickly as possible (within 1500 ms). This served as an attention task. Prior to the random dot motion task, a fixation cross was displayed for 1000 ms. During the random dot motion task, participants had 3000 ms to indicate the direction of the coherent dots. Simultaneously, the advisors indicated the coherent direction. Participants then indicated their confidence level by moving their mouse along a continuous scale. Following each trial, there was a 2000 ms intertrial interval.

On each trial, the participants were first introduced (for 1500 ms) to a trustworthy or untrustworthy advisor (i.e., static image of the deep fake advisor) who would advise in the second phase of the random dot motion task. On 16% of the trials (counterbalanced over trustworthiness and block levels), a red cross was shown just above the nose of the advisors after 250 or 500 ms. Participants had to press the space bar as fast as possible (max 1500 ms). This latter task served the purpose of letting the participants focus on the faces and as an attention check. Next a fixation cross was presented for 1000 ms, after which the second phase started. During the second phase (i.e., Random dot motion task), participants had to indicate the direction of the moving dots, and after 300 ms the advisor, whose face was presented above the random dot task, indicated the direction they thought the moving dots were moving (i.e., by saying 'left' or 'right'). On 50% of the trials the advice was left, and right on the remaining half. Participants had 3000 ms to indicate the direction by pressing the "c" (i.e., Left) or "n" (i.e., Right) key. Following a fixation cross of 1000 ms, the last phase started in which participants had to indicate how confident they were about their decision by moving their mouse on a continuous scale going from 0 (not very confident) to 100 (very confident). They had 10000 ms to decide. After an intertrial interval of 2000 ms, a new trial started. In total this procedure was repeated 96 times, divided over two blocks of each 48 trials. Prior to the experimental phase, participant did a practice run of six trials with a deep fake advisor with a neutral expression. In this practice run, there were four hard trials with correct advice and two easy trials (i.e., 50% left, 50% right).

## Design

Our main measurement of interest was the advice alignment rate, that is whether the advice and the decision of the participants are similar. Additionally measured the choice decision time and confidence ratings. Our experiment was within subject.

## Analyses

All trials in which participants responded faster than 100 ms or that were 2 standard deviations + / -from the participant's mean were considered outliers and removed (5%).

Additionally, when participants did not respond in time during the confidence ratings (> 10000 ms), that trial was considered an outlier and removed from the confidence analyses (.005%).

As written in our preregistration, for our main analyses we only included hard trials with correct advice.

**Main analyses.** For the advice alignment rate, we constructed a general linear mixed model and assumed a binomial distribution with a logistic regression with the "lme4" package [v1.1-34, 71,72]. The fixed effect structure consisted of trustworthiness (i.e., trustworthy, and untrustworthy), and the random effect structure was determined with the backward selection method [73] (See supplementary materials S1 Table). We calculated the p-values with type III Wald Chi-square test, and reported the odds ratio [74] and the asymptotic 95% confidence intervals as the effect size.

For the choice decision time and the confidence ratings, we constructed a linear mixed model with "lmerTest" [v3.1-3, 75] including trustworthiness (i.e., trustworthy, untrustworthy) and advice alignment (i.e., aligned, not aligned) as fixed effects. The random effect structures were determined with the backward selection method [73] (see supplementary materials S1 Table). P-values were calculated with the Sattherwaite (type = III) method in R [v.4.3.1, 61]. The corresponding effect sizes and 95% confidence intervals were calculated with the "effectsize" package [v0.8.6, 76]. To determine the direction of the effects, we ran post hoc t-tests and corrected for multiple comparison with the false discovery rate. We controlled for normality of the residuals by checking the distribution of the residuals and quantile-quantile plots [77].

In addition to the preregistered analyses, equivalence tests were conducted to detect the absence of effects [78,79]. This was done by calculating the overlap of a region of practical equivalence and the 95% confidence intervals for each effect. Equivalence tests help to determine whether the observed effect falls within a region of practical equivalence. In the context of equivalence testing, the null hypothesis posits that the observed effect does not fall within the region of practical equivalence. Conversely the alternative hypothesis asserts that the effects does fall within this region. If the null hypothesis of the equivalence test is rejected, it indicates that the observed data falls entirely within this region, implying that the difference is not practically significant. Conversely, if the null hypothesis cannot be rejected, it suggests that the observed effect lies outside the region of practical equivalence, indicating that the difference may have practical significance [78,79]. We used the bayestestR package [v0.13.1, 80] to determine the region of practical equivalence and ran equivalence tests according to the "TOSTER" recommendations [78,79]. In addition to the p-values related to the hypothesis testing, we report the second-generation p-value (sgpv), a statistic that represents the proportion of data-supported hypotheses that are also null hypotheses [81], for each effect in our main analyses. If this is close to zero, this means that there was almost no overlap between the region of practical equivalence and the 95% confidence interval, and when it is closer to 1 there was almost a 100% overlap.

## Results

### Advice alignment rate

Our analyses revealed that the participants decisions did not align more with the trustworthy (78.3, 95% CI [76.3, 80.1]) advisor compared to the untrustworthy (79.2, 95% CI [77.3, 81.0]) advisor; $\chi^2 1 = 1.44$, $p = .23$ (See Fig 3A; odds ratio: 0.94, 95% CI [0.86, 1.04]). See supplementary materials S2 Table for the model summary. Our equivalence test revealed that the main effect of trustworthiness falls within our region of practical equivalence 95%CI [−0.08, 0.02],

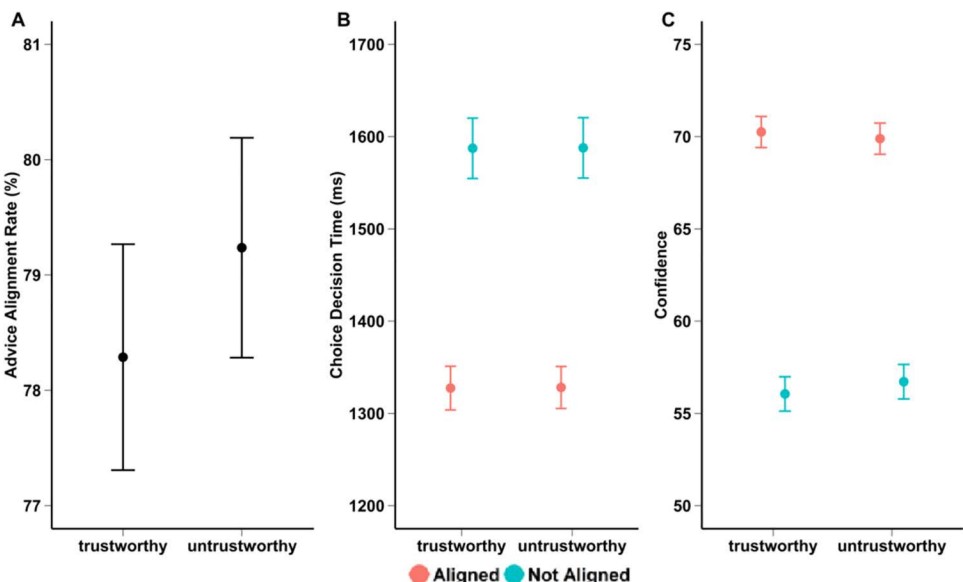

**Fig 3. Graphical depiction results main experiment.** The marginal estimated means for the advice alignment rate (A), choice decision time (B), and confidence ratings (C) (i.e., y-axis) for the second experiment. On the x-axis you can find trustworthiness (i.e., trustworthy, untrustworthy). The graphs for choice decision times and confidence ratings are separated according to following (i.e., red) or not following (blue) the advisor. The black bars (graph A), and red and blue bars (graphs B, C) represent the error rates.

p < .001, sgpv > .999 (regions of practical equivalence = [−0.18, 0.18]), hereby, confirming that the difference in trustworthiness has no practical implications.

## Choice decision time

There was no significant difference in the choice decision time between trustworthy (1457 ms, 95% CI [1405, 1510]) and untrustworthy advisors (1458 ms, 95% CI [1406, 1510]), $F(1, 155.37) = 0.003$, $p = .956$, $\eta_p^2 = 0$ 95% CI [0, 0.00]. Our equivalence test revealed that the main effect of trustworthiness did fall within our region of practical equivalence 95% CI [−10.14, 9.58], p < .001, sgpv > .999 (regions of practical equivalence = [−51.20, 51.20]), confirming that the difference in trustworthiness has no practical implications. However, participants were generally faster when their decision aligned with the advice (1328 ms, 95% CI [1283, 1373]) than when it did not align (1588 ms, 95% CI [1526, 1650]), $F(1, 182.28) = 215.08$, $p < .0001$, $\eta_p^2 = 0.56$ 95% CI [0.46, 0.63]. This was confirmed by our test of equivalence. Indeed, the main of effect of advice alignment [112.52, 147.24] did not fall within the regions of practical equivalence ([−51.20, 51.20]), p > .999, sgpv < .001 Lastly, there was no significant interaction (See Fig 3B) between trustworthiness and advice alignment, $F(1, 155.02) = 0$, $p = .991$, $\eta_p^2 = 0$ 95% CI [0, 0]. See supplementary materials S3 Table for descriptives, and S4 Table for model summary). Our equivalence test revealed that the interaction between trustworthiness and advice alignment did fall with the region of practical equivalence 95% CI [−9.78, 9.98], p < .001, sgpv > .999, confirming that the interaction effect has no practical implications.

## Confidence ratings

Our analyses revealed that there was no significant difference in confidence ratings after deciding when the advisor was trustworthy (62.6, 95% CI [60.8, 64.4]), compared to

untrustworthy (62.9, 95% CI [61.1, 64.7]), $F(1, 10178) = 0.55$, $p = .458$, $\eta_p^2 = 0$ 95% CI [0, 0]. This was confirmed by our equivalence test (region of practical equivalence [−2.04, 2.04]), which showed that the main effect of trustworthiness [−0.50, 0.23], falls within our region of practical equivalence, $p < .001$, sgpv > .999. Participants were, however, more confident when their decision aligned with the advice (70, 95% CI [68.4, 71.6]), compared to when they did not align (55.6, 95% CI [53.4, 57.8]), $F(1, 191) = 300.51$, $p < .0001$, $\eta_p^2 = 0.68$ 95% CI [0.60, 0.74]. Our equivalence tests showed that the main effect of advice alignment ([−8.02, −6.39]), did not fall within the region of practical equivalence $p > .999$, sgpv < .001. However, there was no significant interaction (See Fig 3C) between advice alignment and trustworthiness, $F(1, 10159) = 3.16$, $p = .075$, $\eta_p^2 = 0$ 95% CI [0, 0] (see supplementary materials S5 Table for descriptives, and S6 Table for model summary). Our equivalence tests showed that the inter-action effect ([−0.69, 0.03]) did fall within our region of equivalence, $p < .001$, sgpv > .999. As a result, any difference in the interaction effect between confidence and trustworthiness has no practical implication.

## Exploratory analyses

For exploratory purposes, we conducted three additional analyses. First, we conducted the same analysis as above for advice alignment rate, choice decision time, and confidence ratings, but with difficulty (i.e., easy, hard correct, hard incorrect) as a fixed effect. These analyses were preregistered as exploratory. The random effect structure was determined using the backward selection criterion [73] (See supplementary materials S7 Table for our model comparisons). For further details, we refer to the analyses section of Experiment 2. Please note that we were only interested in the influence of difficulty and will therefore only report the results of these analyses (i.e., main and interaction effects).

Second, we investigated whether participants integrated the advice they received into their decision. This (non-preregistered) analysis was done to rule out the possibility that participants simply ignored the advice and relied solely on their own performance. Specif-ically, in our study, we included both hard trials with correct advice and hard trials with incorrect advice. Given that both trials had the same level of coherence (i.e., 5–10%), the rate of advice alignment should be different for both trial types if participants were influ-enced by the advice they received. To test this, we used a linear mixed model and compared the advice alignment rate in the different difficulty levels (i.e., hard correct, hard incorrect, easy). Please note that, for the sake of completeness, we have also included the easy trials in the analyses. The fixed effect structure consisted of difficulty (i.e., hard correct, hard incorrect, easy), and the random effect structure of (difficulty|subject). Post hoc analyses consisted of the z-ratio and were correct for multiple comparison with the false discovery rate. Please note that for the hard trials with incorrect advice, we inverted the advice align-ment rate as aligning with the advice on such trials would lead the wrong answer. Thus, in the hard trials with incorrect advice, the correct answer was the opposite of the received advice.

Third, we explored the association between participant's confidence and advice align-ment rate. Again, this analysis was not registered. Previous work suggests that confidence is an important predictor of advice following [53]. Therefore, we investigated the impact of confidence on the previous trial on the probability of aligning with the advice on the current trial. We conducted two different analyses: one including only the hard correct trials, and one including all trials. The continuous fixed effect was the scaled confidence ratings on the previous trial, and the random effect was (confidence|subject) for both analyses.

## Exploratory analysis 1: The effect of difficulty

**Advice alignment rate.** When difficulty was included in the model, our analyses showed a significant main effect of difficulty, $\chi^2_2$ = 554.69, $p$ <.0001. Post hoc analyses revealed that the advice alignment rate was higher for easy trials (98.4, 95% CI [98.0, 99.2]), compared to hard correct advice trials (78.8, 95% CI [77.0, 80.2]); $z$ = 14.11, $p$ <.0001, odds ratio = 21.04 95% CI [12.55, 35.30], and hard incorrect advice trials (38.8, 95% CI [35.6, 42.2]), $z$ = 19.20, $p$ <.0001, odds ratio = 123.27, 95% CI [67.63, 224.70]). Likewise, the advice alignment rate was significantly higher for hard trials with correct advice compared to hard trials with incorrect advice, $z$ = 22.09, $p$ <.0001, odds ratio = 5.86, 95% CI [4.84, 7.09]). There was no significant interaction effect between trustworthiness and difficulty (See Fig 4A), $\chi^2_2$ = 2.38, $p$ =.304 (see Supplementary materials S8 Table for descriptives, and S9 Table for the model summary).

**Choice decision time.** When difficulty was included, our analyses revealed that there was a significant main effect of difficulty, F(2, 309.1) = 42.78, $p$ <.0001, $\eta^2_p$ = 0.22 95% CI [0.14, 0.29]. Participants were faster on easy trials (1289 ms, 95% CI [1283, 1341]), compared to hard correct advice trials (1459 ms, 95% CI [1408, 1509]), $t$(911) = −9.24, $p$ <.0001, $\beta$ = −169.28, 95% CI [−213.2, −125.3], and hard incorrect advice trials (1454 ms, 95% CI [1401, 1506]), $t$(516) = −8.30, $p$ <.00001, $\beta$ = −164.57, 95% CI [−212.2, −116.9]. However, there was no significant difference between hard correct advice and hard incorrect advice trials, $t$(182) = 0.49, $p$ =.624, $\beta$ = 4.71, 95% CI [−18.5, 27.9]) (See Fig 4B). Likewise, our analysis revealed a significant interaction effect between difficulty and advice alignment, F(2, 7841.8) = 242.39, $p$ <.0001, $\eta^2_p$ = 0.06 95% CI [0.05, 0.07]. Post hoc analyses revealed that in the easy and hard correct advice conditions, participants were faster when their decision aligned with the advice than when it did not, $t$(2703) = 8.15, $p$ <.0001, $\beta$ = 259, 95% CI [183, 334.5]; $t$(233) = 18.43, $p$ <.0001, $\beta$ = 259, 95% CI [225, 292.6] respectively. However, in the hard incorrect advice trials, participants were faster when their decision did not align with the advice than when it did align, $t$(622) = −5.88, $p$ <.0001, $\beta$ = −105, 95% CI [−148, −62.2] (see Supplementary materials S10 Table for descriptives, and S11 Table for the model summary). Moreover, there was no significant interaction effect between trustworthiness and difficulty, F(2, 16287.3) = 0.32, $p$ =.726, $\eta^2_p$ = 0 95% CI [0, 0], nor between trustworthiness, difficulty, and advice alignment, F(2, 15931.0) = 0.85, $p$ =.426, $\eta^2_p$ = 095% CI [0, 0].

**Confidence ratings.** Our exploratory analyses revealed that participants reported the highest confidence levels on easy (71.1, 95% CI [69.1, 73.1]), compared to hard correct advice (63.3 95% CI [61.7, 64.9]), and hard incorrect advice trials (63.4, 95% CI [61.7, 65.2]), F(2, 358.5) = 41.24, $p$ <.0001, $\eta^2_p$ = 0.19 95% CI [0.12, 0.26]. Post hoc analyses revealed that the confidence ratings

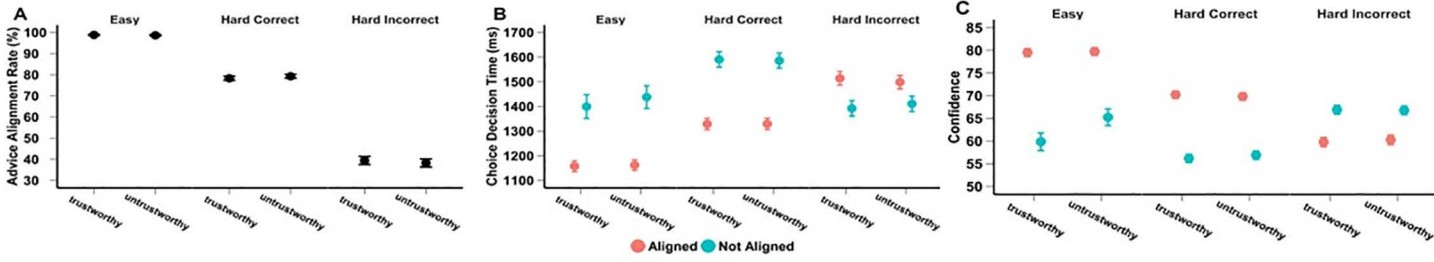

**Fig 4. Graphical depiction results main experiment including difficulty.** *Note.* The marginal estimated mean of the advice alignment rate (A), choice decision times (B), confidence ratings (C) including trial difficulty for the second experiment. On the x-axis you can find levels of trustworthiness (i.e., trustworthy, untrustworthy). The graphs for choice decision times and confidence ratings are separated according to advice alignment (i.e., aligned: red, not aligned: blue), and the trustworthiness (i.e., trustworthy: trustworthy, untrustworthy). The black bars (graph A), and red and blue bars (graphs B, C) represent the error rates.

were significantly higher on easy compared to hard correct advice trials, $t(661) = 9.01$, $p < .0001$, $\beta = 7.81$, 95% CI [5.73, 9.89] and hard incorrect advice trials, $t(445) = 7.72$, $p < .0001$, $\beta = 7.66$, 95% CI [5.27, 10.05]. There was no significant difference between hard correct advice and hard incorrect advice trials, $t = -0.35$, p = .727, $\beta = -0.15$, 95% CI [−1.14, 0.85]. In a similar vein, there was a significant interaction between advice alignment and difficulty (See Fig 4C), $F(2, 14186.5) = 438.80$, $p < .0001$, $\eta_p^2 = 0.06$ 95% CI [0.04, 0.07]. On easy and hard correct advice trials, participants were significantly more confident when they aligned (79.6, 95% CI [78.0, 81.3]; 70.0, 95% CI [68.4, 71.7]), compared to when they did not align with the advice (62.5, 95% CI[59.6, 65.5]; 56.5, 95% CI [54.8, 58.2]), $t(17292) = 13.49$, $p < .0001$, $\beta = -17.08$, 95% CI [−20.11, −14.05]; $t(17485) = 36.79$, $p < .0001$, $\beta = -13.49$, 95% CI [−14.37, −12.61]. However, on hard incorrect advice trials, participants indicated higher confidence levels when their decision did not align to the advice (66.8, 95% CI [65.0, 68.6]), compared to when it did align (60.0, 95% CI [58.1, 61.9]) the advice, $t(9525) = -11.28$, $p < .0001$, $\beta = 6.82$, 95% CI [5.37, 8.26], see supplementary materials S12 Table for descriptives, and S13 Table for model summary). There was no significant interaction between trustworthiness and difficulty $F(2,17484.7) = 2.23$, $p = .108$, $\eta_p^2 = 0$ 95% CI [0, 0.00], nor of advice alignment, task difficulty, and trustworthiness, $F(2,17569.5) = 2.47$, $p = .085$, $\eta_p^2 = 0$ 95% CI [0, 0.00].

Overall, trial difficulty influenced participant's performance. Participants were more likely to align with the advisor, made faster decisions, and reported higher levels of confidence on easy trials compared to hard trials (both correct and incorrect). Although the advice alignment rate was lowest on trials with incorrect advice, there was no significant difference in confidence ratings or choice decision time between hard correct and hard incorrect trials. Furthermore, confidence ratings were higher and choice decision time was faster on trials with correct advice (i.e., easy, hard correct). The opposite effect was found in hard trials with incorrect advice. Participants generally integrated advice into their decisions, especially on trials with correct advice, but rejected incorrect advice (see also Exploratory analysis 2). However, none of these effects were modulated by the trustworthiness of the advisor.

### Exploratory analysis 2: Comparing estimated accuracy

The estimated accuracy was the highest on easy trials (98.7%, 95% CI [98.0, 99.2]) and hard correct trials (78.8, 95% CI [77.0, 80.5]), and the lowest on hard incorrect advice trials (61.2, 95% CI [57.8, 64.4]). Our analyses revealed a significant main effect of difficulty, $\chi_2^2 = 330.1$, $p < .0001$. Post hoc analyses revealed a significant difference in the accuracy between easy and hard correct ($z = 14.08$, $p < .0001$, odds ratio = 21, 95% CI [12.65, 34.85]), between easy and hard incorrect ($z = 17.65$, $p < .0001$, odds ratio = 49.50, 95% CI [29.48, 83.11]. Most importantly, the difference between hard correct and hard incorrect trials was also significant $z = 8.89$, $p < .0001$. The probability of responding accurately was higher for hard trials with correct advice (79%, 95% CI [77,81]) compared to hard trials with incorrect advice (61%, 95% CI [58,64]), odds ratio = 2.36, 95% CI [1.87, 2.97].

The observed difference between hard correct and hard incorrect trials suggests that participants were affected by the advice provided. When inspecting individual advice alignment rates, only 15 out of 199 participants aligned with the advice on more than 70% of the trials, suggesting that the task was sufficiently challenging for most participants and that the observed variation in advice alignment was not due to a ceiling effect.

### Exploratory analysis 3: Confidence and advice alignment rate

Our analyses revealed that when participants were more confident on the previous trial, they were less likely to follow the advice on the current trial. However, the effect was not significant

for hard trials, $\chi^2_1 = 2.40$, $p = .122$, $\beta = -0.05$, 95% CI [$-0.11$, 0.01], and not significant when including all trials, $\chi^2_1 = 0.54$, $p = .464$, $\beta = -0.02$, 95% CI [$-0.06$, 0.03]. These non-significant effects could be attributed to the limited number of trials and the randomisation of trial difficulty.

## General discussion

In this study, we examined the effect of perceived trustworthiness on perceptual decision-making. We successfully created animated versions of computer-generated faces that were previously evaluated as trustworthy or untrustworthy [4]. We applied deep fake models to these static images, and these were combined with audio to produce trustworthy and untrustworthy faces that could give verbal advice. We validated this novel procedure in Experiment 1, and our results showed that trustworthy faces were rated significantly more trustworthy than neutral or untrustworthy faces. Additionally, neutral faces did not significantly differ from the midpoint of our ratings, while trustworthy faces were rated significantly above it, and untrustworthy faces were rated significantly below it. Thus, applying deep fake models to computer-generated faces maintains the trustworthiness properties of the faces [4,7,12,82]. We believe that this novel approach offers researchers the opportunity to create interactive, animated, and more socially engaging versions of established facial databases, potentially enabling a more ecologically valid methodology in the study of social cognition and first impressions.

In Experiment 2, we applied this methodology to create a new set of advisors, which were used as 'trustworthy' and 'untrustworthy' interaction partners in a decision-making task. Our exploratory control analyses suggested that participants integrated the advice in their decision-making process. However, contrary to our predictions, there was no reliable influence of perceived trustworthiness on the advice alignment rate, confidence ratings, or speed with which participants made their decision. This failure to reject the null hypothesis was also confirmed by our equivalence test for all main effects of trustworthiness (i.e., for advice alignment rate, choice decision time, and confidence ratings) and interactions between trustworthiness and advice alignment (i.e., for choice decision time, and confidence ratings).

Participants reported higher confidence ratings and made decisions more quickly when their decisions aligned with the advice, compared to when they did not, regardless of the trustworthiness of these deep fake advisors. For the confidence ratings, this is in line with studies indicating that there is a confidence boost when using advice/information from multiple sources [58]. Similarly, our findings for choice decision times align with previous research, which demonstrated that participants made faster decisions when their choices matched the advice compared to when they did not [18]. Alternatively, this can be explained by task accuracy. Given that we only looked at hard trials with correct advice, it is also possible that participants were more confident because their decision was correct.

The lack of effect for trustworthiness is surprising, given the substantial evidence that perceived trustworthiness influences economic choices [17,83], criminal judgements [21], risk taking behaviour [19], consumer decisions on online platforms [84], and loan decision from creditors [85]. Previous research demonstrated that perceptual trustworthiness also has an impact on advice taking in a value-based decision-making context [18]. We did not find such effects here. However, in the studies conducted by [18], participants engaged in a two-choice decision-making task where they selected one of two doors to find the door hiding a reward. These tasks are considered subjective as there are multiple options and there is no objectively correct answer. In the current study, the task is objective, and there is only one correct answer (i.e., the coherent movement is either to the left or to the right). In their meta-analysis [34], found that individuals are more receptive to advice in subjective tasks with multiple answers [52,53,83] compared to objective tasks, potentially explaining the difference between the

present study and previous work. In a similar vein, participants could rely on their own skills to complete the task. It is therefore possible that participants simply did not pay attention to the social characteristics of the advisors or did not even look at the advisors at all, at least when deciding on the direction of the moving dots. For example, in earlier work showing social biases towards perceived trustworthy faces, the only cues the participants could rely on were the faces and feedback following their decision [18]. Even more, the provided advice was not auditory, but the advisors moved to the option they deemed the best. Here, there was an initial bias to follow perceived trustworthy advisors more, but this effect disappeared throughout the experiment.

An alternative interpretation for the null effect is that perceived trustworthiness is simply not a predictor of advice quality within a perceptual decision-making context. Research has shown that the influence of facial features, such as trustworthiness, is domain specific (For a review see [7]). For instance, when it comes to elections, competence is a significant predictor [86,87], whereas in a military setting [88], facial dominance is an important factor for career advancement. In the context of perceptual decision-making, earlier work demonstrated that the task reputation (i.e., very good at the task, or very bad at the task) of our interaction partner as indicated by visual cues (i.e., 1 star = low reputation, 3 stars = high reputation), influences advice utilisation [46]. Crucially, task reputation was also reflected in the actual accuracy of the advisor (i.e., high reputation – more accurate, low reputation – less accurate). Our paradigm was conceptually similar to [46], but our findings differ from theirs. In our experiment, the advice accuracy was identical for both trustworthy and untrustworthy advisors, meaning that perceptual trustworthiness was not directly indicative of advice accuracy. It is possible that participants recognised this lack of connection. Consequently, our results provide further evidence that social biases are more likely to emerge only when there is a direct association between the social trait and the quality of the advice—specifically, the accuracy of the advisors. This appears to hold true, at least, in an objective task context [34].

Lastly, it is possible that our advice alignment measurement was not sensitive enough to detect potentially small effects of perceived trustworthiness. While we were able to measure when the decision of the participants aligned with the advice, how fast participants decided, and how confident participants were after deciding, it was not possible to directly measure to what extent participants changed their opinion based on the advice [34,89,90]. One paradigm, allowing such measures, is the judge-advisor system [34,36,39,90]. Future studies could integrate deep fake manipulations with advice taking paradigms such as the judge-advisor system to further explore social biases within an advice taking context. Moreover, one could make the task more cooperative, for example by providing a reward for team efforts [91,92], or in general a reward for a correct answer [58]. This may encourage participants to engage more with the advisor and focus more closely on their social characteristics.

Overall, our findings demonstrate, that unlike value-based decision-making [18], perceptual decision making does not seem to be modulated by the perceived trustworthiness of the advisors. This highlights that social biases such as perceived trustworthiness, are potentially domain-specific [7], and that their influence depends on the task requirements [34].

## Conclusion

Advice is an essential and interactive aspect of our decision-making processes in our daily life. However, the impact of the advisor's social characteristics on advice-taking, especially in objective task contexts, is not well comprehended. In this study, we have successfully developed a new procedure to create dynamic trustworthy and untrustworthy faces. In a subsequent step, we examined the impact of perceptual trustworthiness of these faces on perceptual decision-making. Although we observed an effect of advice alignment on confidence ratings

and choice decision times, we did not find any modulation by the trustworthiness of the advisor. We hypothesise that this may be due to the absence of a correlation between the trustworthiness characteristics and the quality of the advice, the domain specificity of perceptual trustworthiness, or task requirements.

## Supporting information

**S1 Table. Models experiment 2.**
(DOCX)

**S2 Table. Model summary for the advice alignment rate (logit).**
(DOCX)

**S3 Table. Descriptives estimated marginal means choice decision time.**
(DOCX)

**S4 Table. Model summary for the choice decision time.**
(DOCX)

**S5 Table. Descriptives estimated marginal means confidence ratings.**
(DOCX)

**S6 Table. Model summary for the confidence ratings.**
(DOCX)

**S7 Table. Models experiment 2 including difficulty.**
(DOCX)

**S8 Table. Descriptives estimated marginal means advice alignment rate including difficulty.**
(DOCX)

**S9 Table. Model summary for advice alignment including difficulty (logit).**
(DOCX)

**S10 Table. Descriptives estimated marginal means choice decision time including difficulty.**
(DOCX)

**S11 Table. Model summary for the choice decision time including difficulty.**
(DOCX)

**S12 Table. Descriptives estimated marginal means confidence ratings including difficulty.**
(DOCX)

**S13 Table. Model summary for confidence ratings including difficulty.**
(DOCX)

## Acknowledgments

We used DeepL Write, and ChatGPT 4o for a spelling and language check. We thank Doctor Zhang Chen for proofreading the manuscript.

## Author contributions

**Conceptualization:** Mathias Van der Biest, Sam Verschooren, Frederick Verbruggen, Marcel Brass.

**Data curation:** Mathias Van der Biest.

**Formal analysis:** Mathias Van der Biest.

**Funding acquisition:** Mathias Van der Biest.

**Investigation:** Mathias Van der Biest.

**Methodology:** Mathias Van der Biest, Sam Verschooren, Frederick Verbruggen, Marcel Brass.

**Project administration:** Mathias Van der Biest.

**Software:** Mathias Van der Biest.

**Supervision:** Sam Verschooren, Frederick Verbruggen, Marcel Brass.

**Validation:** Mathias Van der Biest, Sam Verschooren, Frederick Verbruggen, Marcel Brass.

**Visualization:** Mathias Van der Biest.

**Writing – original draft:** Mathias Van der Biest.

**Writing – review & editing:** Sam Verschooren, Frederick Verbruggen, Marcel Brass.

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
