## [Decision Letter · Decision Letter 0]

5 Nov 2024

PONE-D-24-19564Perceptual judgments are resistant to the advisor's perceived level of trustworthiness: a deep fake approachPLOS ONE

Dear Dr. Brass,

Thank you for submitting your manuscript to PLOS ONE. After careful consideration, we feel that it has merit but does not fully meet PLOS ONE’s publication criteria as it currently stands. Therefore, we invite you to submit a revised version of the manuscript that addresses the points raised during the review process.

Both reviewers raised concerns about the methods and writing. Specifically, Reviewer #2 highlighted important issues regarding experimental design and data analysis that should be addressed in your revision. You may want to consider running a control experiment without providing any advice to establish baseline performance

We look forward to receiving your revised manuscript.

Kind regards,

Alireza Soltani

Academic Editor

PLOS ONE

Journal requirements: When submitting your revision, we need you to address these additional requirements. 1. Please ensure that your manuscript meets PLOS ONE's style requirements, including those for file naming. The PLOS ONE style templates can be found at https://journals.plos.org/plosone/s/file?id=wjVg/PLOSOne_formatting_sample_main_body.pdf and https://journals.plos.org/plosone/s/file?id=ba62/PLOSOne_formatting_sample_title_authors_affiliations.pdf 2. Thank you for stating the following financial disclosure:  [FundingM.V.D.B. (11K2721N) and S.V (1212721N) are supported by the Research Foundation Flanders. F. V is supported by an ERC Consolidator grant (European Union’s Horizon 2020 research and innovation programme, Grant Agreement No 769595). M.B. is supported by an Einstein Strategic Professorship of the Einstein Foundation Berlin (EPP-2018-483) and by the Deutsche Forschungsgemeinschaft (DFG, German Research Foundation) under Germany's Excellence Strategy–EXC 2002/1 ‘Science of Intelligence’–project number 390523135.].  Please state what role the funders took in the study.  If the funders had no role, please state: ""The funders had no role in study design, data collection and analysis, decision to publish, or preparation of the manuscript."" If this statement is not correct you must amend it as needed. Please include this amended Role of Funder statement in your cover letter; we will change the online submission form on your behalf. 3. Please include your full ethics statement in the ‘Methods’ section of your manuscript file. In your statement, please include the full name of the IRB or ethics committee who approved or waived your study, as well as whether or not you obtained informed written or verbal consent. If consent was waived for your study, please include this information in your statement as well. 4. Your ethics statement should only appear in the Methods section of your manuscript. If your ethics statement is written in any section besides the Methods, please move it to the Methods section and delete it from any other section. Please ensure that your ethics statement is included in your manuscript, as the ethics statement entered into the online submission form will not be published alongside your manuscript.  5. We notice that your supplementary tables are included in the manuscript file. Please remove them and upload them with the file type 'Supporting Information'. Please ensure that each Supporting Information file has a legend listed in the manuscript after the references list. 6. Please include a caption for figure 1, 2, and 3A.

Reviewers' comments:

Reviewer's Responses to Questions

**Comments to the Author**

1. Is the manuscript technically sound, and do the data support the conclusions?

Reviewer #1: Partly

Reviewer #2: Partly

2. Has the statistical analysis been performed appropriately and rigorously? 

Reviewer #1: Yes

Reviewer #2: No

3. Have the authors made all data underlying the findings in their manuscript fully available?

Reviewer #1: Yes

Reviewer #2: Yes

4. Is the manuscript presented in an intelligible fashion and written in standard English?

Reviewer #1: Yes

Reviewer #2: Yes

5. Review Comments to the Author

Reviewer #1: This study investigates how trustworthiness of social information affects perceptual judgements in human decision-making. They operationalize trustworthiness through deep fake generated faces/ They show that trustworthiness does not affect the judgment.

The study asks an interesting question, but I am unsure how valid trustworthiness here is. As the authors say in the Discussion that trustworthiness in the study is completely unassociated with the task at hand. I would say that it is unassociated with anything really, except for facial features. This connects to my biggest problem with the study – why are facial features used at all to operationalize trustworthiness?? What is the logic behind it? I suspect racist and sexist biases influence which facial features are deemed to be trustworthy or not.

I like the Discussion – it addresses the limitations of the study quite well. A simple extension of this experiment would be to cultivate trustworthiness of advisers through either clearly showing initially whether the advice given is correct and incorrect by a particular person and then testing how people incorporate the advice based on trustworthiness. Or if the focus is on perceptual trustworthiness, then avtars that speak or gesture more confidently could be used to imply trustworthiness.

Can the authors show the different faces used for trustworthiness? I think there are 4?

Minor point: Discussion lines 467-469 need to be rewritten for better sentence structure.

Reviewer #2: Research question and methods:

In this manuscript, Van der Biest et al. ask whether the trustworthiness of an advice giver impacts the advice taking rate of advice receiver on perceptual judgments. To that end, they conduct a random dot motion discrimination experiment where individuals have to indicate whether a patch of randomly moving dots move to right or left direction while they alter the percentage of dots that move coherently (coherence level). Crucially, this coherence level is adjusted to create two different levels of task difficulty as the authors hypothesize that the level of uncertainty in the environment interacts with trustworthiness and affects the advice taking behavior.

Although the question is very interesting and the study has merits, there are critical issues in the methodology and the statistical analysis that authors need to address in order for the results of the study to be clear. I will describe these issues in the following.

Methods:

- The authors measure the advice taking rate as the frequency of the trials that the participant has adhered to the decision of another person. This has a confound with participant’s own decision because the authors cannot tease apart the decision of the participant, in the absence of any advice, from the decision made by following the advice. In other words, it is not clear whether a participant’s decision was made by their own judgment or was influenced by the advice giver. Authors should connect additional data with which there are trials or blocks where the participants do not receive any advice, and they have to make decisions based on their own judgment. This creates a baseline to compare the own’s decision with the decision to adhering to the advice.

- I wonder why the design of the experiment is in a way that all trials with easy coherence involve correct advice instead of having both correct and wrong advice in all coherence levels.

- I am not sure about the rationale behind picking up the coherent levels in the RDM task, but 55%-75% of motion coherence is usually considered very high and make the task very easy. Usually, values between 0 and 24% are selected for the motion coherence level because participants usually reach a very high performance level around 24% coherence level. I think the performance is essential to check and report in the two conditions because it might be the case that the performances are very high in this task (due to picking up the 55%-75% coherence) and therefore, the desired interaction that authors expected between trustworthiness and uncertainty has not been observed (because people are usually more susceptible to the opinion of others when there is uncertainty in the environment).

Technical issues:

- The axis label needs to be changed whenever it says “advice following rate” because due to the same issue mentioned above, the authors cannot tease apart choice following advice from one’s own decision.

- What is the accuracy rate of participants? I think it is essential to have a violin plot of the accuracies with individual dots representing the accuracy of each individual and to make sure that the participants do not reach a ceiling level of accuracy.

- In supplementary materials, the analyses have been separated into easy, hard, and incorrect advice. While it is clear from the text that the incorrect advice happen during the hard trials only, it is necessary to put the labels correctly (i.e., easy correct, hard correct, hard incorrect).

- Is the RT in the main text the average of RTs across the three conditions in the supplementary materials? If yes, why do we see an opposite trend in most of the trials (easy and hard) in the supplementary materials?

- While it is interesting to use deep fake technology to come up with more realistic social stimuli, the faces of the advice givers in the experiment are not natural, making the participants less susceptible to be influenced by them. Thus, I am not sure about the real use of the deep fake here.

- The flow of the manuscript in the introduction seems like a post-hoc analysis at some parts. For example, the authors hypothesize that they would expect to see and interaction between trustworthiness and advice following for decision time while they do not mention anything about why they expect this interaction in the literature review.

Statistical analyses:

- The authors have used linear mixed models to predict advice taking rate, but they have not done any model comparisons. For example, why the confidence rate is not a variable in the linear mixed model (as the authors also mention in the introduction that it is a variable that influences the advice taking)? The same model comparison issue also applies to the confidence and response time modeling.

- It would also be interesting to look at the effect of current trials’ confidence on the next trial advice taking rate as the advice taking might show itself on the confidence history of the participants.

- The authors mention that they only focused on the hard trials with correct advice for their analysis, but I wonder why they did not consider hard trials with wrong advice too as they also could be the high uncertain conditions (even maybe more than correct trials, because correct trials are already in alignment with participant’s decision).

Writing:

- In general, the manuscript has room for improvement in the writing. I recommend the authors to revise the manuscript for clarity.

- This is minor but I believe that the format that the current introduction has been written could be changed to convey less redundant information, and to make it clearer what each paragraph is talking about. As an example (but not just limited to this one), the last paragraph begins with redundant information with the previous paragraph which could be moved there.

6. PLOS authors have the option to publish the peer review history of their article (what does this mean? ). If published, this will include your full peer review and any attached files.

**Do you want your identity to be public for this peer review?** For information about this choice, including consent withdrawal, please see our Privacy Policy .

Reviewer #1: No

Reviewer #2: **Yes: ** Aryan Yazdanpanah

---

## [Author Response · Author response to Decision Letter 1]

13 Dec 2024

We have modified the manuscript in accordance with the PLOS ONE guidelines.

[Funding

M.V.D.B. (11K2721N) and S.V (1212721N) are supported by the Research Foundation Flanders. F. V is supported by an ERC Consolidator grant (European Union’s Horizon 2020 research and innovation programme, Grant Agreement No 769595). M.B. is supported by an Einstein Strategic Professorship of the Einstein Foundation Berlin (EPP-2018-483) and by the Deutsche Forschungsgemeinschaft (DFG, German Research Foundation) under Germany's Excellence Strategy–EXC 2002/1 ‘Science of Intelligence’–project number 390523135.].

We have included a statement in the funding section.

We have included a comprehensive ethics statement in the Methods section, along with an explanation of how participants provided their consent.

The ethics statement is included in the method section.

5. We notice that your supplementary tables are included in the manuscript file. Please remove them and upload them with the file type 'Supporting Information'. Please ensure that each Supporting Information file has a legend listed in the manuscript after the references list.

We have uploaded all supporting tables as separates files, and wrote a legend which can be found after the references list.

6. Please include a caption for figure 1, 2, and 3A.

We have included a caption for all figures.

Reviewer #1: This study investigates how trustworthiness of social information affects perceptual judgements in human decision-making. They operationalize trustworthiness through deep fake generated faces/ They show that trustworthiness does not affect the judgment.

The study asks an interesting question, but I am unsure how valid trustworthiness here is. As the authors say in the Discussion that trustworthiness in the study is completely unassociated with the task at hand. I would say that it is unassociated with anything really, except for facial features. This connects to my biggest problem with the study – why are facial features used at all to operationalize trustworthiness?? What is the logic behind it? I suspect racist and sexist biases influence which facial features are deemed to be trustworthy or not.

Thank you for your insights. The facial stimuli used in this study were derived from a validated model designed to create faces that vary in perceived trustworthiness in a statistical and holistic manner. To the best of our knowledge, these social perceptions are not influenced by racist or sexist biases. Instead, research suggests that humans tend to overgeneralize subtle emotional expressions (Zebrowitz & Montepare, 2008), particularly in uncertain contexts (Oosterhof & Todorov, 2008), which are evolutionarily linked to approach and avoidance responses (Todorov, 2008; Todorov et al., 2015). Faces with positive valence are typically perceived as trustworthy, whereas those with negative valence are perceived as untrustworthy. This concept has been elaborated in the introduction (lines 116–129).

I like the Discussion – it addresses the limitations of the study quite well. A simple extension of this experiment would be to cultivate trustworthiness of advisers through either clearly showing initially whether the advice given is correct and incorrect by a particular person and then testing how people incorporate the advice based on trustworthiness. Or if the focus is on perceptual trustworthiness, then avtars that speak or gesture more confidently could be used to imply trustworthiness.

Thank you for this suggestion. Previous research has shown that manipulating the validity of advisors significantly affects advice following rates in both value-based (Van der Biest et al., 2020) and perceptual decision-making (Qi et al., 2018). While these studies demonstrate that knowledge-based trustworthiness manipulations influence decision-making, they cannot rule out the possibility that these effects are non-social, reflecting cue-following behavior rather than trustworthiness per se. When advice is consistently accurate, the optimal heuristic is simply to follow it and disregard inaccurate advice, regardless of social impressions. In the present study, we used perceptual manipulations to investigate how first impressions based on trustworthiness shape advice-following behavior without manipulating the validity of the received advice.

We agree that future research could explore the integration of multiple sources of perceptual trustworthiness cues, such as eye gaze duration, vocal tone, and gestures associated with confidence.

Can the authors show the different faces used for trustworthiness? I think there are 4?

Yes, these can be found on OSF (see line 197 and OSF).

Minor point: Discussion lines 467-469 need to be rewritten for better sentence structure.

We have rewritten this section.

Reviewer #2: Research question and methods:

In this manuscript, Van der Biest et al. ask whether the trustworthiness of an advice giver impacts the advice taking rate of advice receiver on perceptual judgments. To that end, they conduct a random dot motion discrimination experiment where individuals have to indicate whether a patch of randomly moving dots move to right or left direction while they alter the percentage of dots that move coherently (coherence level). Crucially, this coherence level is adjusted to create two different levels of task difficulty as the authors hypothesize that the level of uncertainty in the environment interacts with trustworthiness and affects the advice taking behavior.

Although the question is very interesting and the study has merits, there are critical issues in the methodology and the statistical analysis that authors need to address in order for the results of the study to be clear. I will describe these issues in the following.

Methods:

- The authors measure the advice taking rate as the frequency of the trials that the participant has adhered to the decision of another person. This has a confound with participant’s own decision because the authors cannot tease apart the decision of the participant, in the absence of any advice, from the decision made by following the advice. In other words, it is not clear whether a participant’s decision was made by their own judgment or was influenced by the advice giver. Authors should connect additional data with which there are trials or blocks where the participants do not receive any advice, and they have to make decisions based on their own judgment. This creates a baseline to compare the own’s decision with the decision to adhering to the advice.

Thank you for this insight. The paradigm we employed is conceptually similar to that of Qi et al. (2018; see lines 110-112, 301-303, 477–485), who investigated how task reputation influences perceptual decision-making. Their findings demonstrated that task reputation affects whose advice participants follow or ignore. The goal of our experiment was to show differences in advice following rates when receiving advice from perceived trustworthy versus untrustworthy advisors, independent of participants' actual accuracy or overall advice following rates.

We agree that it is important to confirm that participants utilize the provided advice, but we argue that an additional experiment is unnecessary. In our study, we included both hard trials with correct advice and hard trials with incorrect advice. Given that both trials have the same coherency levels (i.e., 5-10%), the advice alignment rate should be the same (see our comments on your suggestion to change the term advice following rate). However, if the advice alignment rate, is different this would suggest that they are influenced by the advice that they receive. In other words, if participants are not influenced by the advice received, their performance on hard trials should be equally good or bad, regardless of whether the advice was correct or incorrect.

Therefore, we statistically investigate whether the advice alignment rate was different in these two conditions (i.e., hard correct advice vs hard incorrect advice). To test this, we used a linear mixed model and compared the advice alignment rate in both types of trials (i.e., hard correct, and hard incorrect). Please note that only for the hard trials with incorrect advice, we inverted the advice alignment rate. As aligning with the advice is in this case the wrong answer. Thus, in the hard trials with incorrect advice, the correct answer was the opposite of the received advice.

Our results demonstrated that the advice alignment rate was significantly different in both types of trials, which suggests that participants did use the advice they received (χ²2 = 331.17, p < .0001). The probability of aligning with the advisor was 2.36 times, CI [1.87,2.97] higher during hard trials with correct advice (79%, 95% CI [77, 81]) compared to hard trials with incorrect advice (61%, 95% CI [58, 64]).

This significant difference suggests that participants are influenced by the advice that they receive. However, this is not influenced by the perceived trustworthiness of the advisor.

- I wonder why the design of the experiment is in a way that all trials with easy coherence involve correct advice instead of having both correct and wrong advice in all coherence levels.

If advisors made mistakes on easy trials, this would likely have affected their perceived trustworthiness. Specifically, in easy trials where the direction of dot movement is clear and there is little to no uncertainty, providing incorrect advice would influence both the advice-following rate (Van der Biest et al., 2024) and perceived trustworthiness (Chang et al., 2010). We have included this motivation in the manuscript (lines 300 - 304).

- I am not sure about the rationale behind picking up the coherent levels in the RDM task, but 55%-75% of motion coherence is usually considered very high and make the task very easy. Usually, values between 0 and 24% are selected for the motion coherence level because participants usually reach a very high performance level around 24% coherence level.

We thank the reviewer for highlighting this point. In our study, we adopted the terminology of the ROK-plugin (Strittmatter et al., 2023), where a coherency of 50% indicates that the stimuli are ambiguous (e.g., 50% of the dots move to the right, and 50% to the left). Consequently, a coherency of 55% means that 5% of the dots are coherently moving in one direction (left or right). Our coherency levels align closely with the literature and the reviewer’s suggestions. Specifically, for hard trials, the coherency ranged from 5–10%, while for easy trials, it was set at 25%.

We have revised the manuscript to incorporate the terminology recommended by the reviewer (e.g. , 142, 299).

I think the performance is essential to check and report in the two conditions because it might be the case that the performances are very high in this task (due to picking up the 55%-75% coherence) and therefore, the desired interaction that authors expected between trustworthiness and uncertainty has not been observed (because people are usually more susceptible to the opinion of others when there is uncertainty in the environment).

Building on our response to the previous comment, we argue that task difficulty was not the reason for the absence of an interaction. For hard trials, the coherency ranged between 5–10%, and our data showed that on hard trials with incorrect advice, participant accuracy (influenced by the advice) was only 61%. This indicates that the task difficulty was consistent with prior research and sufficiently challenging to investigate the hypothesized interaction.

Technical issues:

- The axis label needs to be changed whenever it says “advice following rate” because due to the same issue mentioned above, the authors cannot tease apart choice following advice from one’s own decision.

Thank you for this suggestion. We have changed the term advice following rate to advice alignment rate. We have changed this throughout the manuscript, appendix, and supplementary materials.

- What is the accuracy rate of participants? I think it is essential to have a violin plot of the accuracies with individual dots representing the accuracy of each individual and to make sure that the participants do not reach a ceiling level of accuracy.

We estimated participant accuracy based on performance in the hard trials with incorrect advice, where accuracy was approximately 61%, which is most likely deflated by the received advice. For the hard trials with correct and easy trials, the estimated accuracy is most likely inflated by the received advice.

To visualize the data, we created a violin plot displaying individual data points for advice alignment rates across easy trials, hard trials with correct advice, and hard trials with incorrect advice.

Fig 1. Violin Plot of the Advice Alignment Rate (%) by difficulty

Note. Graphical depiction of the individual advice alignment rates for each participant. On the x-axis you can find the advice alignment rate, and on the y-axis the different difficulty levels (i.e., easy, hard correct, and hard incorrect). Each dot represents an individual participant. Especially for the hard correct trials, we can see that the data is following a normal distribution.

Upon closer inspection of the data, only 15 out of 199 participants agreed with the advice on more than 70% of trials, indicating no ceiling effect for participant alignment rate (or accuracy). Additionally, we clarified in the manuscript that the coherency in the hard trials was 5–10% (see response to the previous comment), which made the task sufficiently challenging. As a result, a ceiling effect is unlikely.

- In supplementary materials, the analyses have been separated into easy, hard, and incorrect advice. While it is clear from the text that the incorrect advice happen during the hard trials only, it is necessary to put the labels correctly (i.e., easy correct, hard correct, hard incorrect).

Thank you for this suggestion. We have changed the labels of the figures in the appendix, as well as in the tables and the text.

- Is the RT in the main text the average of RTs across the three conditions in the supplementary materials? If yes, why do we see an opposite trend in most of the trials (easy and hard) in the supplementary materials?

All reported results in the manuscript are the marginal estimated means according to the respective model for the hard trials with correct advice. In the appendix, we repeated the analyses with difficulty as a factor. However, we have realised that the legen

---

## [Decision Letter · Decision Letter 1]

14 Jan 2025

PONE-D-24-19564R1Perceptual judgments are resistant to the advisor's perceived level of trustworthiness: a deep fake approachPLOS ONE

Dear Dr. Brass,

Thank you for submitting your manuscript to PLOS ONE. After careful consideration, we feel that it has merit but does not fully meet PLOS ONE’s publication criteria as it currently stands. Therefore, we invite you to submit a revised version of the manuscript that addresses the points raised during the review process.

**Please note that your revision will not be sent out for external review but will instead be evaluated at the editorial level.**

**Specifically, we believe that the results of some of your analyses highlighted in response to Reviewer #2 are critical for understanding and appreciating the main findings and should be incorporated into the main manuscript. These include: 1. Figure 1 from the Appendix and the accompanying text should be included in the main manuscript as Figure 4 (though the tables can remain in the Appendix). 2. The argument against a ceiling effect (from your rebuttal, starting with "Upon closer inspection of the data...") should also be included in the main text. Related to the this, the accuracy rate for different conditions should be reported in the main text. 3 . The results on alignment rate as a function of confidence rating from previous trials are interesting and should be reported in the main text, even if the results are not statistically significant.**

We look forward to receiving your revised manuscript.

Kind regards,

Alireza Soltani

Academic Editor

PLOS ONE

**Journal Requirements:**

Reviewers' comments:

Reviewer's Responses to Questions

**Comments to the Author**

1. If the authors have adequately addressed your comments raised in a previous round of review and you feel that this manuscript is now acceptable for publication, you may indicate that here to bypass the “Comments to the Author” section, enter your conflict of interest statement in the “Confidential to Editor” section, and submit your "Accept" recommendation.

Reviewer #1: All comments have been addressed

Reviewer #2: All comments have been addressed

2. Is the manuscript technically sound, and do the data support the conclusions?

Reviewer #1: Yes

Reviewer #2: Yes

3. Has the statistical analysis been performed appropriately and rigorously? 

Reviewer #1: Yes

Reviewer #2: Yes

4. Have the authors made all data underlying the findings in their manuscript fully available?

Reviewer #1: Yes

Reviewer #2: Yes

5. Is the manuscript presented in an intelligible fashion and written in standard English?

Reviewer #1: Yes

Reviewer #2: Yes

6. Review Comments to the Author

**Reviewer #1: ** (No Response)

**Reviewer #2:**  I think the authors have addressed the raised issues sufficiently, and I wish the authors good luck and a happy new year!

7. PLOS authors have the option to publish the peer review history of their article (what does this mean? ). If published, this will include your full peer review and any attached files.

**Do you want your identity to be public for this peer review?** For information about this choice, including consent withdrawal, please see our Privacy Policy .

Reviewer #1: No

Reviewer #2: **Yes: ** Aryan Yazdanpanah

---

## [Author Response · Author response to Decision Letter 2]

17 Jan 2025

Specifically, we believe that the results of some of your analyses highlighted in response to Reviewer #2 are critical for understanding and appreciating the main findings and should be incorporated into the main manuscript. These include: 1. Figure 1 from the Appendix and the accompanying text should be included in the main manuscript as Figure 4 (though the tables can remain in the Appendix).

We have incorporated the Appendix into the manuscript, including the results section and figure (Lines 422-532). However, please note that our focus has been on the main and interaction effects involving difficulty, as these are the primary analyses of interest. Consequently, we have not included the main effects of trustworthiness or advice alignment rate, nor their interaction in the manuscript nor in the figure (only figures A-C are included). Additionally, the tables from the Appendix have been moved to the supplementary materials, and as a result, the Appendix can now be removed from the manuscript.

"2. The argument against a ceiling effect (from your rebuttal, starting with "Upon closer inspection of the data...") should also be included in the main text. Related to the this, the accuracy rate for different conditions should be reported in the main text.

We incorporated this section directly following our difficulty analyses (Lines 512 - 526). Please note that the results in the manuscript (χ²2 = 330.1, p < .0001) slightly deviate from the results reported in our previous response letter (χ²2 = 331.17, p < .0001). There were no differences in the marginal estimated means, nor did this change the interpretation of the results. We discuss this control analyses in the general discussion (Lines 548 - 549).

3 . The results on alignment rate as a function of confidence rating from previous trials are interesting and should be reported in the main text, even if the results are not statistically significant.

Thank you for this suggestion. We have reanalyzed the data, including all trials as well as only hard trials. The z, and p-value (χ²1 = 2.40, p = .122) slightly deviates from our previous response letter (χ²1 = 2.11, p = .147). There was no difference in the marginal estimated means. Furthermore, this does not change the interpretation of the results. We have incorporated these analyses into the manuscript (Lines 527 -532).

---

## [Editor Report · Decision Letter 2]

26 Jan 2025

Perceptual judgments are resistant to the advisor's perceived level of trustworthiness: a deep fake approach

PONE-D-24-19564R2

Dear Dr. Brass,

We’re pleased to inform you that your manuscript has been judged scientifically suitable for publication and will be formally accepted for publication once it meets all outstanding technical requirements.

Kind regards,

Alireza Soltani

Academic Editor

PLOS ONE
---

## [Editor Report · Acceptance letter]

PONE-D-24-19564R2

PLOS ONE

Dear Dr. Brass,

I'm pleased to inform you that your manuscript has been deemed suitable for publication in PLOS ONE. Congratulations! Your manuscript is now being handed over to our production team.

Kind regards,

on behalf of

Dr. Alireza Soltani

Academic Editor

PLOS ONE